# "I'm suffering for food": Food insecurity and access to social protection for TB patients and their households in Cape Town, South Africa

**Lieve Vanleeuw**[1,2☯]*, **Wanga Zembe-Mkabile**[1,3☯], **Salla Atkins**[2,4‡]

**1** Health Systems Research unit, South African Medical Research Council, Tygerberg, South Africa, **2** New Social Research and Global Health and Development, Faculty of Social Sciences, Tampere University, Tampere, Finland, **3** Archie Mafeje Social Policy Research Institute, School of Transdisciplinary Research and Graduate, Studies, University of South Africa, Pretoria, South Africa, **4** Department of Global Public Health, Karolinska Institutet, Stockholm, Sweden

☯ These authors contributed equally to this work.
‡ SA also contributed equally to this work.
* lieve.vanleeuw@mrc.ac.za

**Data Availability Statement:** Data underlying the results presented in this article cannot be shared publicly as study participants did not give consent to share the anonymised data publicly after the

## Abstract

### Background

Tuberculosis (TB) is a major health concern and the number one cause of death in South Africa. Social protection programmes can strengthen the resilience of TB patients, their families and households. This study aimed to get a better understanding of the role of social protection and other forms of support in relation to the burden of TB on patients and their households in South Africa.

### Methods

This is a cross-sectional exploratory qualitative study using a phenomenological approach to focus on the lived experiences and perceptions of TB patients and healthcare workers. We interviewed 16 patients and six healthcare workers and analysed data thematically.

### Results

The challenges faced by participants were closely related to household challenges. Participants reported a heavy physical burden, aggravated by a lack of nutritious food and that households could not provide the food they needed. Some needed to resort to charity. At the same time, households were significantly affected by the burden of caring for the patient—and remained the main source of financial, emotional and physical support. Participants reported challenges and costs associated with the application process and high levels of discretion by the assessing doctor allowing doctors' opinions and beliefs to influence their assessment.

### Conclusion

Access to adequate nutritious food was a key issue for many patients and this need strained already stretched households and budgets. Few participants reported obtaining state social

study. The original application to the ethics committee did not include an explicit request to participants to share data. Anonymised data is available after reasonable request to the SAMRC Ethics Committee for study replication. Please find the contact details for the SAMRC Ethics Office: Research Integrity Office Secretariat, Ms A Labuschagne (E-mail: Adri.Labuschagne@mrc.ac.za).

**Funding:** LV received internal funding from the South African Medical Research Council for this study. The funder provided support in the form of salaries for authors LV and WZ but did not have any additional role in the study design, data collection and analysis, decision to publish, or preparation of the manuscript. The specific roles of these authors are articulated in the 'author contributions' section.

**Competing interests:** The authors have declared that no competing interests exist.

**Abbreviations:** AFSUN, African Food Security Network; DG, Disability Grant; DoL, Department of Labor; DOT, directly observed therapy; DR-TB, drug resistant tuberculosis; DS-TB, drug sensitive tuberculosis; HIV, human immunodeficiency virus; PHC, primary health care; SAMRC, South African Medical Research Council; SASSA, South African Social Security Agency; TB, Tuberculosis; UIF, Unemployment Insurance Fund; WHO, World Health Organisation.

protection support during their illness, but many reported challenges and high costs of trying to access it. Further research should be conducted on support mechanisms and interventions for TB patients, but also their households, including food support, social protection and contact tracing. In deciding eligibility for grants, the situation of the household should be considered in addition to the individual patient.

## Introduction

Tuberculosis (TB) claims more than a million lives each year and affects millions more, with enormous impacts on families and communities [1]. In South Africa, TB is the number one cause of death [2]. The recently released First National TB Prevalence Survey South Africa 2018 found that TB rates in South Africa have been underestimated. In 2018, an estimated 390,000 people fell ill with TB as compared to the previous estimate of 301,000 cases, and, despite available treatment, an estimated 63.000 people died. The survey also reported over 150,000 undiagnosed and untreated TB cases [3].

In South Africa, and elsewhere, those from lower socioeconomic strata are more at risk of developing active TB, and risk becoming poorer as a consequence [4, 5]. TB is a key example of the "poverty disease trap", inextricably linked with social determinants of health which are all largely shaped by income poverty including food scarcity and undernutrition, HIV, overcrowding, poor living conditions, drug use and lack of social support [6]. TB also worsens poverty through medical and non-medical costs (direct costs), and loss of income by reducing patients' physical strength and ability to work (indirect costs), posing a significant economic burden on affected households [7–10]. A systematic review of the financial burden of tuberculosis faced by patients and affected families in low- and middle-income countries found that the total cost of being ill with TB was equivalent to 58% of reported annual individual and 39% of reported household income. On average, 20% of the total cost was due to direct medical costs, 20% to direct non-medical costs, and 60% to income loss [10]. In South Africa, the unemployment rate of TB patients has been found higher (54%) than the general population (30%), but direct costs of treatment and care were low as treatment is free. The expenses incurred by patients were nevertheless catastrophic as many patients reported no income. Patients reported coping strategies such as selling assets and borrowing money, which increased over time as financial losses depleted household and social resources [8].

TB is in essence a community disease, with a high transmission rate [11, 12]. It affects families and households, not only individuals. Ill-health and poverty can erode a family's resilience to withstand and bounce back from adversity. An accumulation of multiple, persistent challenges, as with chronic illness or disability, conditions of poverty, or ongoing, complex trauma can overwhelm the family, heightening vulnerability and risk for subsequent problems [13]. Social protection programmes can strengthen families' resilience by reducing vulnerability and complementing coping strategies. The World Health Organisation (WHO) recognizes the bidirectional relationship link between poverty and TB and the impact of both on families and households. The WHO End TB strategy has explicitly included a target that no TB affected families should face catastrophic costs (defined as total costs of TB diagnosis and care above 20% of the household's annual income) [14, 15] due to TB. The End TB strategy recommends that governments use universal health coverage and social protection and interventions to reduce poverty, ensure food security and improve living and working conditions to reduce TB incidence and mortality [14]. Social protection in the form of cash grants has been shown to

positively impact TB risk factors [16] reduce out-of-pocket payments [17], improve treatment completion and reduce default rates for TB patients [18].

South Africa, having adapted the End TB strategy and having a well-established social welfare system has the potential and the mechanisms to provide social protection to TB patients and their families. TB patients can apply for state provided social assistance in the form of the Disability Grant (DG), provided to people with a physical or mental disability that are unfit to work for a minimum of six months. Studies indicate, however, that only 5% of drug sensitive TB patients and 44% of drug-resistant TB patients accessed the DG grant [8, 19].

Despite multiple studies confirming the link between TB and poverty in South Africa [8, 9, 20, 21], and several studies reporting on the benefits of the DG for illnesses such as HIV [22–28], few studies in South Africa have explored access to state provided social grants for TB patients [8, 19]. These studies were quantitative and did not provide detail on the implications of the burden of TB on patients and their households, or the role of and access to social protection and other forms of support to alleviate that burden. Our qualitative study aimed to get a better understanding of the role of social protection and other forms of support in relation to the burden of TB on patients and their households through interviews with TB patients and healthcare workers in Cape Town, South Africa.

## Methodology

### Study design

We conducted a cross-sectional exploratory study using a phenomenological approach through qualitative interviews, to focus on the lived experiences and perceptions of TB patients and healthcare workers.

### Context and setting

The study was conducted in 2017–2018 in two primary healthcare (PHC) clinics in one of the oldest townships on the outskirts of Cape Town, Western Cape, South Africa. The township has a high population density (16958 persons/km2), with more than 40% of inhabitants living in informal dwellings such as shacks, and 22.2% of the working age population reporting no income [29]. The Western Cape Province had the third highest TB incidence in South Africa with 681 cases per 100 000 population in 2015, well above the national average of 520 cases per 100 000 population. In the same year, Cape Town had the second highest TB burden in South Africa with 23 815 cases [30]. PHC clinics provide treatment and care for drug-sensitive TB (DS-TB) and drug-resistant TB (DR-TB) free of charge. The City of Cape Town has more than 130 community-based clinics, ensuring healthcare services close to patients' homes. Both study clinics have a dedicated "TB room" that diagnoses, initiates on treatment, and provides Directly Observed Therapy (DOT).

### Recruitment and sampling

We aimed to interview TB patients and healthcare professionals treating TB patients in two clinics. A convenience sample of TB patients was drawn from TB patients attending the clinic for diagnosis treatment and treatment reviews. The research assistant approached patients in the waiting area of the TB room and asked if they would like to participate in the study. When the patient agreed to participate, a pre-interview participant profile form was filled with the patient. The information collected included patients' socioeconomic characteristics, TB history in the household, and receipt of social grants. The study team approached the nurses and doctors working in the TB room and asked if they would like to participate in the study.

## Data collection

Semi-structured interviews were conducted with 16 TB patients and six healthcare workers (two doctors and four nurses) at both clinics. Interviews with TB patients were conducted in Xhosa (S1 File) or English (S2 File), depending on the patient's preference. Interviews with healthcare workers were conducted in English (S3 File). The interviews with TB patients explored how patients experienced having TB, how it affected their life and that of the household they lived in, their experience of health care services, and experience of applying for the DG. The interviews with healthcare workers explored experiences and perceptions of healthcare workers on challenges for TB patients, support needed, and the process of applying for the DG. All interviews were recorded, transcribed verbatim, translated into English and checked against the original recording to ensure accuracy.

Following each day of interviews, LV and WZ discussed the interviews, noting initial thoughts and meanings, and emerging themes related to the original research question and considering new areas for further interviews. Transcripts and notes were deidentified and stored on South African Medical Research Council's (SAMRC) secured servers.

## Data analysis

Transcripts and notes were analysed using thematic content analysis [31] to identify and interpret patterns and themes in the qualitative data. Transcripts were read and re-read to allow for familiarisation and to start the process of open coding. Coding was performed inductively on MS Excel. Quotes were interpreted and condensed for meaning, then organized into sub-categories and categories (Table 1). Preliminary analysis was performed by LV and reviewed by WZ and SA following which the analysis was revised.

## Researcher characteristics and reflexivity

The study was conducted by three researchers, WZ, LV and SA. All have masters' qualifications in social sciences, with WZ and SA doctoral training in public health research. All were experienced in qualitative research. WZ is a black Xhosa-speaking female, LV is a white foreign English-speaking female and SA a white foreign English speaking female, living away from South Africa. WZ and LV conducted the interviews and SA participated in analysis.

WZ understood that she brought with her the insider-outsider perspective and experience to each interview and to the research process as a whole, as someone who both embodies the lived experience of Black low-income South Africans, but who, through her educational qualifications and her current middle class positionality in the socio-economic strata, is removed

**Table 1. Example of coding.**

| Quote | code | Condensed meaning | Sub-category | category |
|---|---|---|---|---|
| it was evening while asleep in my flat I got uncomfortably hot and I threw off the blankets and then even had to take my clothes off; at this stage my breathing was really straining I could hardly breathe. I went inside the house to wake up my family, they organised me a car to take me to XX Hospital. When I got there I was made to blow on something (take a breathalyser? And then I was sent to YY Hospital. It was at YY where they diagnosed me with MDR. | 2 months after DS-TB Tx course, developed severe symptoms (strained breathing) | severe symptoms | very ill | Burden: patient |
| | family organised car to take her to XX clinic | | family support | Support: patient |
| | was transferred from vanguard clinic to YY hospital | hospitalised for MDR | hospitalised | Burden: patient |
| | diagnosed with MDR at YY | hospitalised for MDR | hospitalised | Burden: patient |

from the immediate economic conditions and experiences of participants residing in a township. WZ used this awareness to forge a connection with respondents and ensured that it did not cloud the data collection process.

LV was aware that being white and foreign, despite long residence in South Africa, might be received with feelings of suspicion or resentment linked to the country's history of apartheid and colonialisation. LV therefore spent time building rapport with healthcare workers to gain their trust. Both researchers ensured that enough time was spent explaining the study as well as answering any questions regarding the study or the researchers' background to create transparency and reduce anxiety.

SA brought with her a background of living and working in South Africa for 14 years but having lived away for over 10 years. Her work in South Africa focused on evaluating TB services. She has strong views on equity, social justice and believes in the potential of social protection systems to support health globally.

### Ethics approval and consent to participate

The study received approval from the Human Research Ethics Committee at the South African Medical Research Council (EC015-82017, October 2017). All participants were given informed consent forms which were read together with the participant and explained in detail before forms were signed. Participants were informed about the purpose of the study, procedures involved, risks and benefits of the study and their rights as participants. The right to decline participation was emphasised, as well as an assurance given that the decision (not) to participate would not affect the healthcare service received at the clinic or repercussions from clinic staff. Participants were given an assurance of confidentiality and strict protection of collected data. The participant characteristics are aggregated to protect participant identities.

## Results

Between November 2017 and February 2018, semi-structured interviews were conducted with 16 patients and six healthcare workers (Table 2).

The healthcare workers included four nurses and two doctors of which five were female and one was male. Nurses had between one to five years' experience working with TB patients, whereas both doctors had been working with TB patients for more than five years. Four of the healthcare workers were less than 40 years old, while 2 were older.

**Table 2. Details of TB patients in the study.**

|  | Female | Male |
|---|---|---|
| **Gender** | 10 | 6 |
| **DS-TB** | 7 | 3 |
| **DR-TB** | 3 | 3 |
| **HIV positive** | 10 | 4 |
| **Employed** | 1 | 1 |
| **Unemployed** | 9 | 5 |
| **Receives DG** | 2 | 1 |
| **Age range** |  |  |
| 20–30 years old | 5 | 2 |
| 31–40 years old | 4 | 3 |
| 40 years and older | 1 | 1 |

We identified three overarching themes: 1) the burden of TB on the individual, 2) family support and the burden of TB on the household, and 3) limited access to social protection.

## The burden of TB on the individual

Participants in the study spoke at length of the heavy physical burden of both TB disease and anti-TB treatment. Before diagnosis and start of treatment, most participants reported symptoms of night sweats, weight loss, coughing, tight chest, difficulty breathing, and appetite loss. For many participants, however, the time between onset of symptoms and diagnosis of TB was long, i.e. several weeks to several months, resulting in many of them becoming so ill they had to be admitted to hospital.

> "Those early days I was weak I could not even walk on my own, I was always in bed. When I needed to walk around the house for instance or use the bathroom; I would have to hold on to walls until I get to where I needed to go."

> (TB patient, DR-TB, female)

> "I was losing weight, and not eating. I was always sleeping, tired, and my limbs were weak, and couldn't move."

> (TB patient, DS-TB, male)

Reasons for the long time between symptoms and diagnosis were on both the patient and healthcare system side. Patient-related reasons were mostly an underestimation of the cough, thinking it was a flu and treating it with over-the-counter medication. On the side of the health system, about half of the participants mentioned the clinic taking a long time to diagnose. Close to half of the participants mentioned the clinic performing several tests, mostly X-rays, that would return negative. Two participants with a delayed diagnosis had TB of the stomach and TB meningitis respectively which could explain the negative results. Reasons for delayed care-seeking remain an area that requires further investigation

> "I would keep coughing, and kept coming but they said that they couldn't see it. I would think but I'm dying, I can feel I'm really coughing, I'm sweating."

> (TB-patient, DS-TB, female)

While symptoms cleared quickly after starting treatment, most patients in our study, both DS-TB and DR-TB, reported debilitating treatment side-effects. The most common side-effects were vomiting and nausea after taking medication, especially when treatment was taken without food. Several patients reported that the medication made them feel so lethargic that they needed to sleep immediately after taking treatment. Many of them also complained of swollen legs and painful feet since being on treatment, which made their daily clinic walk in the first weeks of treatment difficult.

> "My feet were killing me. . .Even though they were painful I needed to go to the clinic every day. . . I would force myself to walk. . . I would take old socks and tie them tight around them and would walk, this way I would not feel that much pain."

> (TB patient, DR-TB, female)

For many participants the physical burden of TB was closely linked to food. Participants reported a change in their dietary needs, most notably an increased need for fruit and vegetables, but also red meat, as recommended by their doctor. One of the doctors described how TB patients, especially those that are very sick, require increased food to support recovery:

"Once they get sick it becomes all that more important to get that well balance diet but it costs so much more. We have to take that into consideration when they first start their treatment especially when they have lost a lot of weight and are very weak. So they have to actually change their diet and add calories so that they have to gain weight."

(TB doctor)

Participants also reported a need for certain foods to minimize the nausea and vomiting caused by the medication such as porridge, juice and yoghurt.

"Taking treatment, it's been very painful because you ingest it. Having ingested it, there is something that it does in your body. It can be harmful, especially when there is nothing in your stomach."

(TB patient, DS-TB, male)

More nutritious food, however, came with a cost that most participants couldn't afford. Only two participants were employed, while three reported earning a small income before they fell ill with TB from informal trading or informal jobs, which stopped once they fell ill. All but two participants lived in households where food is purchased and prepared for the entire household. To deal with the extra cost, patients reported having to employ strategies to make the available money and groceries for the household last.

"When we get cash I buy a tray or two but I make sure it lasts me until Friday when we get money again. I control my cravings and not plough into the food at once."

(TB patient, DS-TB, female)

Others attested that the household groceries simply could not cater for their illness. Worst off, however, were two participants who did not live with family and struggled daily to buy the most basic food but also reported frequently missing meals.

"There are times there would be no food, sometimes we go to sleep without eating... Perhaps you eat once a day, you struggle to take pills because there is no food."

(TB patient, DS-TB, female)

One of the doctors confirmed that "The main concern to most patients is just food to eat" (TB doctor).

The doctors also explained that doctors at the PHC clinic can refer TB patients that are considered underweight (BMI <18.5kgs) for food support packages. However, the TB doctor admitted that the food support packages were often not enough.

## Family support and the burden on the household

As already suggested above, the individual burden of TB was closely connected to a household burden of TB. Most participants lived in a household with a large extended family.

Several participants reported living in crowded conditions where family members shared a room. Most also knew someone in their close environment with TB that could have possibly infected them: a family member, boyfriend or girlfriend, or neighbor. While the presence of TB in many families, combined with crowded living conditions, increases the risk of transmission, most patients reported that no contact screening was performed in their household and that adult members of the household (roommates, boy/girlfriends, neighbours or backyard dwellers) were not requested to go to TB screening. Only one participant, who had several family members with TB and got infected while caring for her brother who was sick with TB, reported that the clinic asked her to bring members of her household to screening at the clinic. Patients that lived with small children were asked to bring the children under 5 years old.

The shared household, while a transmission risk, was a key source of support for many unemployed participants. The support provided by family members was mainly related to food, including buying groceries and cooking meals. But patients also mentioned family members walking them to the clinic, taking over chores at home, looking after the patient's children, and bathing and clothing the patient. Several patients mentioned a family member that traveled from the Eastern Cape to Cape Town to come look after them.

Two participants, however, were alone in Cape Town and had no family support, having migrated from the Eastern Cape to seek work. Both participants lived hand to mouth from informal jobs, struggled daily and regularly did not have enough money to buy food. One, a 23-year-old woman, told a harrowing story of being ill with TB with no family support and no income. She lived in a room that was shared with another couple and braided hair for a living. She was diagnosed with both HIV and TB at hospital. Following three weeks of hospitalization she was discharged but still felt weak and had painful feet. Yet she walked to the clinic every day to collect her medication. To make an income, she would braid hair for as little as R100 ($6) during which she needed to stand on her painful feet. She bought groceries with the little money she made. She finally turned to community charity when she ran out of money.

Patients without family support found the lack of physical and emotional care especially difficult.

"But I mean I was really suffering a lot when I was- was sick . . . I was really suffering because there was no one taking care of me at home in the first place. There's no one who's concerned if I've had anything to eat, and what I need. . . You can't have someone make you porridge, who are you going to ask, who's going to make it for you?"

(TB patient, DS-TB, female)

A doctor at one of the clinics confirms that those patients with family support tend to do better than others:

"But I have found that patients with some kind of support are likely to do better."

(TB doctor)

Family support, however, does not come without its own challenges both for the family and the patient. Patients talked about the sacrifices the family had to make to care for them. As mentioned earlier, for example, two participants' family had travelled from the Eastern Cape to care for them. Another patient attested how she left her job to look after her brother with TB, who eventually infected her, too. Patients also talked about the strain their illness puts on

an already stretched family, especially families without a regular income. Several patients also spoke of the physical and mental burden on the household from caring for a patient with TB.

> "That's another thing with living in a household with people and having TB, is that the understanding of the dynamics, they just—anybody gets sick and tired of tending after a person for a while."

(TB patient, DS-TB, female)

## Limited access to social protection

In South Africa, social protection in the form of a Disability Grant (DG) or food parcels, also called Social Relief of Distress (SRD), can support patients in the absence of family but also support families to continue caring for patients. None of the participants in our study, however, received the Social Relief od Distress or food parcel, and one of the doctors in the study confirmed that

> "In our experience we find that patients really don't get that—Sassa always run out of funds to be able to help patients who require that social relief assistance."

(TB doctor)

In terms of the DG, of the 16 patients interviewed, only three received this grant, ten were in the process of applying, and three of them had not applied. Of those that received the DG, two had DR-TB and one had DS-TB combined with HIV, diabetes and high blood pressure. Participants generally had little information about the application and assessment process for the DG and reported that they heard about the grant and application process from other TB patients and from family and neighbours. Information about the DG was not generally available in the clinic and participants had to explicitly ask and wait for the doctor or social worker to share the information. Nurses participating in the study said they had no knowledge of the application process and referred patients to the doctor or social worker. Doctors in the study confirmed that they are the main point of information and assistance with the DG application process in the clinic and that this takes up a considerable amount of their time. For those participants that had applied for the DG, the application process was a long and arduous process with many back and forth between the South African Social Security Agency (SASSA) office and the clinic, but also to their previous employer, the Department of Labor (DoL), Home Affairs, the bank, and the police, to obtain documents required for the application.

> "It was a most difficult task my sister, there is a lot of leg work involved. . .I left here with the letter the Doctor drafted and went to South African Social Services Agency who had a form for the Doctor to complete and then I needed to take the form back to SASSA and I had to go my bank. I may have gone at least four to five times to South African Social Services Agency."

(TB patient, DR-TB, male)

According to participants, the long process starts with a referral letter given by the doctor that is taken to the SASSA office who in return gives the patient forms that the doctor must complete. In addition to the medical assessment, the patient needs to submit a host of personal documentation such as certified copies of the identity document, bank statements, proof of income, proof of assets, proof of marital status, proof of residence, statement from the

Unemployment Insurance Fund or discharge certificate from the previous employer. The main obstacle for patients who were previously employed was to obtain evidence that they were no longer employed and did not receive a financial benefit from the Unemployment Insurance Fund (UIF). Participants suggested that obtaining these documents required several visits to the DoL and the previous employer, adding strain and delaying the application process. Participants reported that the run around for the DG application was very costly, mainly because of transportation costs. One patient admitted that she had to use the Child Support Grant of her child to pay for the transport costs.

The medical assessment in Cape Town is performed by a PHC TB doctor. They indicate on the form whether s/he recommends the DG. The medical report must confirm that the patient has a physical or mental disability that limits or prevents them from doing any work in order to maintain themselves. In the absence of a clear guideline or list of conditions that qualify for the disability grant, the doctor must base his/her recommendation on the patient's clinical picture. As one doctor explained:

"You use your discretion, you use your clinical knowledge and expertise in that area and decide whether the patient qualifies for the grant or not."

(TB doctor)

A doctor's 'discretion' however can be based on or influenced by his/her personal feelings and beliefs. One doctor emphatised with patients.

"If we are hoping for patients to stay on the treatment and reducing the defaulter rate, then really I think there should be more people on the disability grant."

(TB doctor)

Another doctor, however, stated that he "wouldn't advise patients, you know, to get money" because

"That money they're not going to use it for food . . . they will go and buy clothes. And they will go and buy cell phones."

(TB doctor)

Participants in the study, however, said they would mostly use the money for food, electricity, rent, and the children's needs.

"I would be able to give myself energy, buy the foods that are compatible with the treatment. Something that will give me a boost so that I'm alright."

(TB patient, DS-TB, male)

Several participants, mostly women, also mentioned that it would give them back their independence.

"The money helped my sister, because they (household members) received grants but it prioritised their own needs. . . My mother and my brothers, they received their respective grants. . . but now I can go out and buy what I need as well."

(TB patient, DS-TB, female)

"I'm going to be able to buy it myself and have my own stash (food) at home, where I say 'okay guys, this cupboard is the sick person's cupboard.'"

(TB patient, DS-TB, female)

## Discussion

Our study reports experiences of TB patients of the burden of TB on themselves and their households, and the different kinds of support, including social protection, available to them. To our knowledge it is the only manuscript to report such data in South Africa.

Most participants reported a heavy physical burden and severe illness before diagnosis, in many cases leading to hospitalization. Patient-related reasons for the delay in diagnosis can be improved with more education and awareness of TB. The facility-related reasons, however, i.e. inability to diagnose timeously, is more concerning and needs further investigation.

Most also reported knowing someone in their close environment with TB and several patients reported living with a previous or current TB patient, adding to the burden on the household. Despite this reality, however, most households of TB patients were not screened for TB, except for children under the age of five years old. Several studies have reported on the low rates of screening and testing for TB by the healthcare system, resulting in missed TB patients and indicating that passive case finding approaches are insufficient [32, 33]. The recently released First National TB prevalence study in South Africa confirms that a concerningly high number of TB cases are missed by the healthcare system. The survey further reports that a high number of TB cases is asymptomatic but also that those with symptoms delay seeking care, resulting in continued transmission in affected households and communities [34]. Actively finding TB patients, especially among close contacts of know TB patients, is essential to break the TB transmission cycle.

The heavy physical burden was aggravated for most patients by a lack of nutritious food. Doctors in the study confirmed that nutritious food is essential to recovery for TB patients. Many participants, however, reported that the household food basket could not provide the extra nutrition that was needed, while others disclosed that they struggled to buy even the most basic food and had to rely on charity. Food insecurity, an insufficiency of food in the household combined with an inability to acquire enough nutritious food, has been previously established among TB patients. The African Food Security Network (AFSUN) 2008–2009 survey of patients with HIV/TB in 11 southern African cities showed that only 10% of patients with TB resided in food-secure households [35]. Food insecurity increases the risk of progression from latent to active disease, is a risk factor for infection, and can negatively impact on adherence [36]. A qualitative study in Swaziland found that some food-insecure patients stopped TB treatment because it increased their appetites when they were already food scarce [37]. Malnutrition has also been shown to increase mortality among TB patients [38]. Conversely, food assistance has been shown to improve adherence and treatment completion [39].

Food insecurity in TB patients in South Africa is driven by high levels of unemployment and poverty in households. While treatment and care for TB are free in South Africa and most patients did not spend on transport to access healthcare, patients reported that the loss of income (formal or informal), the cost of food and the cost of applying for the disability grant, were their main financial stressors. These findings correspond with previous studies that reported substantial direct and indirect costs that TB patients incur leading to households selling assets and borrowing money, placing a significant burden on households and driving the "medical poverty trap" [8, 19]. Further research also suggests that TB- affected households remain economically vulnerable even after TB- treatment completion, with limited recovery in income and employment, and persistent financial strain [40].

While households were significantly affected by the burden of TB disease, they remained the main source of financial, emotional and physical support for most TB patients in our study. Many households were already stretched but stretched themselves even further to be able to provide shelter, food and care for family members with TB. Family support has previously been shown to be an important driver of patient adherence [41–45]. The bidirectional relationship between family and health, whereby the family is affected by the health of its members but also plays an important role in the health of its members, has only more recently been studied [46, 47]. Recurring episodes of TB and persistent financial strain can adversely affect family functioning and decrease the family's resilience, especially in poorer families, making it more vulnerable to shocks and less able to withstand those shocks and rebound. Our study clearly showed the effect of TB on the household and the household's impact on a patient with TB. Moreover, our study showed that TB patients without family support struggled to access food daily and were in a constant state of survival. More research is needed on the impact of TB on family functioning and resilience, including interviews with household members, and on interventions such as social protection to improve household health and resilience.

State provided support in the form of food parcels or cash grants can enable TB patients to continue their treatment and get healthy again but can also support TB affected households to strengthen their resilience and support their ability to care for household members with TB. While there is limited evidence on the effect of food supplementation and financial support on treatment outcomes for TB patients [48, 49], financial support has been shown to positively impact on TB risk factors [16] increase treatment success [18], improve clinic attendance [49] and reduce out-of-pocket payments for TB patients [17]. In addition, the National Strategic Plan for HIV, TB and STI 2017–2022 recognises the vital role of households in supporting TB patients and the need to empower and strengthen households as key actors. The Department of Social Development (DSD) is to play the lead role in building strong social support systems and has several tools to its disposal to support TB patients and their households such as the Social Relief of Distress (SRD) or food parcel and the Disability Grant (DG). Effectiveness and efficiency of food versus cash has, to our knowledge, not been investigated in the South African context, yet research conducted in sub-Saharan African, Asian and South-American countries has shown that both cash transfers and food aid are effective but cash transfers and vouchers tend to be more efficient and less costly than food-based interventions [50]. None of the participants in our study, however, received the SRD or food parcel, and few were successful in their application for the DG.

The application process for the Disability Grant presented many challenges for TB patients. The first challenge is access to information about the application process. Participants in our study testified that information on the DG was not available at the clinic, nor from the nurses. Patients are routinely referred to the doctor as he/she is the one that assesses the patients and makes a recommendation, severely limiting access to information. A further challenge is the administrative burden or 'leg work' and financial costs involved when applying for the DG which for many can become catastrophic and will have been in vain as only a small fraction of TB patients were awarded the DG. This concurs with previous studies that showed only 5% of TB patients received the DG in South Africa (8). Whilst there appears to be commitment to addressing social determinants of TB, the South African TB programme has historically mainly focused on the medical treatment model with investments in innovative technologies, better treatment, and reorganization of healthcare services. A review of budgetary allocations towards addressing the socio-economic drivers of TB, however, does not reflect the claimed level of commitment [51]. A challenge here is the policy assumption underlying the DG which focuses on medical conditions that cause significant functional loss and limitation of normal

daily activities, thereby excluding illnesses like TB and HIV that can be successfully controlled with medication [25]. The DG, as many income security mechanisms in low- and middle-income countries, is not explicitly designed to provide income support in periods of ill health and often excludes those who have the greatest need [52]. Policies including sickness insurance or a basic income grant could address this gap.

Equally concerning, however, was the finding that despite many regulatory changes to reduce the discretion of the doctor [25], our study showed that doctors performing the medical assessment for the application continue to have high levels of discretion i.e. doctors decide if a person is sick enough and needy enough to "deserve" social assistance, allowing doctors' opinions and beliefs such as misuse of grants to influence their assessment. Despite perceptions of misuse of grants such as the Disability Grant, Child Support Grant and the Old Age Grant, there is no evidence supporting this. Social grants, however, have been shown to play an important welfare function in poor households as they are typically shared [53].

High levels of discretion in assessments for social grants has previously been shown to be a critical issue in assessing people living with HIV to receive the disability grant [54]. The Harmonised Assessment Tool (HAT), developed and piloted jointly by the Department of Social Development and the Department of Health, introduced a more holistic and standardized assessment for the DG, reducing the discretion of the doctor, but has never been implemented.

More research on support mechanisms and interventions for TB patients and their households, including food support and social protection, is recommended. In addition, a review of the application process for the disability grant is highly recommended.

## Strengths and limitations

Our study was explorative and conducted with TB patients attending two clinics in Cape Town. The results can therefore not be generalized to all TB patients. Our findings, however, highlight issues likely to be relevant to many TB patients, especially those living in similar socio-economic circumstances. A second limitation is that the interviews were conducted in 2017–2018, nearly 4 years ago. Nevertheless, the challenges presented in this article remain relevant and have to date not been resolved nor addressed. In addition, there has been no change in the social grants policy for TB patients in South Africa. Lastly, while interviews with TB patients revealed the burden on their households, our study did not interview household members which could have complemented patients' experiences of the burden on the household. Further research by the authors is investigating the burden of TB on households.

## Conclusion

Our study highlights the co-relation between TB patients and their households in a context of high unemployment, poverty and lack of food security. Patients that did not have family support struggled more—but those that had support reported their households and families struggling, too. Access to adequate nutritious food was a key issue for many patients. However, their illness placed further strain on already stretched households and budgets. Few participants reported obtaining state social protection support during their illness, but many reported high costs of trying to access it. Social protection could be key to protecting the resilience of patients, and their households against financial shock in times of illness. Further research should be conducted on support mechanisms and interventions for TB patients, but also their households, including food support, social protection and contact tracing. In deciding eligibility for grants, the situation of the household should be considered in addition to the individual patient. Finally, a review of the application process for the disability grant is recommended.

## Supporting information

**S1 File. Topic guide for TB patients—Xhosa.**
(PDF)

**S2 File. Topic guide for TB patients—English.**
(PDF)

**S3 File. Topic guide for healthcare workers—English.**
(PDF)

## Author Contributions

**Conceptualization:** Lieve Vanleeuw, Wanga Zembe-Mkabile, Salla Atkins.

**Data curation:** Lieve Vanleeuw.

**Formal analysis:** Lieve Vanleeuw.

**Funding acquisition:** Lieve Vanleeuw.

**Investigation:** Lieve Vanleeuw.

**Methodology:** Lieve Vanleeuw.

**Project administration:** Lieve Vanleeuw.

**Writing – original draft:** Lieve Vanleeuw.

**Writing – review & editing:** Wanga Zembe-Mkabile, Salla Atkins.

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
