## [Decision Letter · Decision Letter 0]

19 Oct 2021

PONE-D-21-11255“I’M SUFFERING FOR FOOD”: FOOD INSECURITY AND ACCESS TO SOCIAL PROTECTION FOR TB PATIENTS AND THEIR HOUSEHOLDS IN CAPE TOWN, SOUTH AFRICAPLOS ONE

Dear Dr. Vanleeuw,

Thank you for submitting your manuscript to PLOS ONE. After careful consideration, we feel that it has merit but does not fully meet PLOS ONE’s publication criteria as it currently stands. Therefore, we invite you to submit a revised version of the manuscript that addresses the points raised during the review process.

We look forward to receiving your revised manuscript.

Kind regards,

Andrew Medina-Marino, PhD

Academic Editor

PLOS ONE

Journal Requirements:

3. Please include a copy of the interview guide used in the study, in both the original language and English, as Supporting Information, or include a citation if it has been published previously.

4. Thank you for stating the following financial disclosure: "LV received funding from the South African Medical Research Council for this study. www.samrc.ac.za. The funders had no role in study design, data collection and analysis, decision to publish, or preparation of the manuscript."

We note that one or more of the authors is affiliated with the funding organization, indicating the funder may have had some role in the design, data collection, analysis or preparation of your manuscript for publication; in other words, the funder played an indirect role through the participation of the co-authors. If the funding organization did not play a role in the study design, data collection and analysis, decision to publish, or preparation of the manuscript and only provided financial support in the form of authors' salaries and/or research materials, please do the following:

a. Review your statements relating to the author contributions, and ensure you have specifically and accurately indicated the role(s) that these authors had in your study. These amendments should be made in the online form.

b. Confirm in your cover letter that you agree with the following statement, and we will change the online submission form on your behalf: 

“The funder provided support in the form of salaries for authors [insert relevant initials], but did not have any additional role in the study design, data collection and analysis, decision to publish, or preparation of the manuscript. The specific roles of these authors are articulated in the ‘author contributions’ section.

Reviewers' comments:

Reviewer's Responses to Questions

**Comments to the Author**

1. Is the manuscript technically sound, and do the data support the conclusions?

Reviewer #1: Yes

Reviewer #2: Yes

2. Has the statistical analysis been performed appropriately and rigorously? 

Reviewer #1: N/A

Reviewer #2: N/A

3. Have the authors made all data underlying the findings in their manuscript fully available?

Reviewer #1: Yes

Reviewer #2: Yes

4. Is the manuscript presented in an intelligible fashion and written in standard English?

Reviewer #1: Yes

Reviewer #2: Yes

5. Review Comments to the Author

Reviewer #1: The goal of this qualitative study of TB patients in South Africa was to describe role for social protection programs for TB patients and access to them. The authors have done a wonderful job describing some critical issues.

They describe the deblitation of TB patients at the time of diagnosis and discuss the need for active, rather than passive, case finding. Next, they discuss the food insecurity and undernutrition prevalent among persons with TB. Lastly, they discuss the convoluted and prohibitively expensive road to a disability grant.

I had three additional thoughts for the authors to consider:

1. Wish the authors could add a little material regarding why the patient’s had delayed getting care. Was it fear of income loss? Inability to get to the health center easily? This is relevant to social protections for high risk populations and would help target active case finding programs.

2. The idea that the health care workers were not prepared to guide the patients towards disability grants suggests a real need to ensure that TB clinics and hospitals take on the role of helping patients navigate the arduous task of obtaining grants available to them. This is an important entrypoint for intervention. How did the healthcare workers interviewed feel about this? Did they feel that it was their responsibility or did they need administrative support for this?

3. Could the authors speak to the validity of the TB doctors concern that disability grant funds would be diverted to luxury items rather than nutrition. Did the use of the disability grant come up in their interviews with the patients?

Overall, I agree with the conclusions made based on the presented data and applaud this paper.

Reviewer #2: This is a very interesting research and adds value to the body of evidence for social determinants of health, especially TB. Authors have done a good job of exploring the perspectives of doctors as well as the patients. The methodology is sound.

I have a few suggestions for strengthening the discussion:

1. A little more on cash vs. food support will be useful in putting the things into perspective for future researchers and advocacy. Any studies that have explored that in South Africa will be great value. The common arguments are of misuse of cash (as mentioned by the doctor) or food sharing in case of food support (the recurring theme is food in many quotes).

2. A point in discussion to emphasize the support to family rather than just the patient. Especially considering the renewed focus on households in National Strategic Plan of South Africa for HIV, TB and STIs 2017-2022.

3. This was done in 2017-18 and we are now in 2021. It will be a good idea to put this limitation upfront and mention the newer developments or the lack of it, in SDoH/social protection with reference to TB in the country.

4. Discuss the findings of this study and lack of evidence for support for patients (nutrition and others) as found by 10.1002/14651858.CD007952.pub3; 10.1002/14651858.CD006086.pub4

6. PLOS authors have the option to publish the peer review history of their article (what does this mean?). If published, this will include your full peer review and any attached files.

Reviewer #1: No

Reviewer #2: **Yes: **Madhavi Bhargava

---

## [Author Response · Author response to Decision Letter 0]

11 Dec 2021

Dear Reviewers, 

thank you for the helpful and constructive comments that have allowed us to improve the paper for publication. 

Please find our point by point response below. We also include a copy of the revised manuscript with tracked changes to illustrate the changes made, as well as a clean copy.

We hope that these changes have much improved the paper.

REVIEWER 1

1. Wish the authors could add a little material regarding why the patient’s had delayed getting care. Was it fear of income los? Inability to get to the health center easily? This is relevant to social protections for high risk populations and would help target active case finding programs. Dear reviewer, thank you for this comment. We looked at the patients’ interviews data again and have made additions to the results and discussion section regarding reasons for the long time between onset of symptoms and diagnosis.

Changes/Additions made to results (line 195-199 and 206-214): 

For many participants, however, the time between onset of symptoms and diagnosis of TB was long, i.e. several weeks to several months, resulting in many of them becoming so ill they had to be admitted to hospital. 

Reasons for the long time between symptoms and diagnosis were on both the patient and healthcare system side. Patient-related reasons were mostly an underestimation of the cough, thinking it was a flu and treating it with over-the-counter medication. On the side of the health system, about half of the participants mentioned the clinic taking a long time to diagnose. Close to half of the participants mentioned the clinic performing several tests, mostly X-rays, that would return negative. Two participants with a delayed diagnosis had TB of the stomach and TB meningitis respectively which could explain the negative results. Reasons for delayed care-seeking remain an area that requires further investigation 

“I would keep coughing and kept coming but they said that they couldn’t see it. I would think but I'm dying, I can feel I'm really coughing, I'm sweating.” (TB-patient, DS-TB, female)

Addition made to discussion (line 367-369): 

Patient-related reasons for the delay in diagnosis can be improved with more education and awareness of TB. The facility-related reasons, however, i.e. inability to diagnose timeously, is more concerning and needs further investigation.

2. The idea that the health care workers were not prepared to guide the patients towards disability grants suggests a real need to ensure that TB clinics and hospitals take on the role of helping patients navigate the arduous task of obtaining grants available to them. This is an important entry point for intervention. How did the healthcare workers interviewed feel about this? Did they feel that it was their responsibility or did they need administrative support for this? Dear reviewer, thank you for this comment. We looked at the healthcare worker interviews again and made changes and additions to the results and discussion section regarding their perceptions and experiences of assisting TB patients with the application process for social grants. However, we disagree that this is an important entry point for intervention. Patient knowledge about the grant application process and requirements would not help much on its own because access to the grant is doctor-driven. Doctors decide if a person is sick enough and needy enough to ‘’deserve’’ social assistance (line 344-367; line 454-462). We feel that the decision to grant social assistance should not be based on doctors’ discretion. It should be informed by a holistic assessment including other actors involved in TB patient care such as social workers. The Harmonised Tool suggested by DSD a few years ago which never took flight, was a step in this direction. We have added a few lines emphasizing this point in the discussion. 

Changes/Additions made to results section (line 309-317): 

Participants generally had little information about the application and assessment process for the DG and reported that they heard about the grant and application process from other TB patients and from family and neighbours. Information about the DG was not generally available in the clinic and participants had to explicitly ask and wait for the doctor or social worker to share the information. Nurses participating in the study said they had no knowledge of the application process and referred patients to the doctor or social worker. Doctors in the study confirmed that they are the main point of information and assistance with the DG application process in the clinic and that this takes up a considerable amount of their time. 

Changes/addition to discussion (line 431-437): 

None of the participants in our study, however, received the SRD or food parcel, and few were successful in their application for the DG. 

The application process for the Disability Grant presented many challenges for TB patients. The first challenge is access to information about the application process. Participants in our study testified that information on the DG was not available at the clinic, nor from the nurses. Patients are routinely referred to the doctor as he/she is the one that assesses the patients and makes a recommendation, severely limiting access to information.

Addition to discussion (line 452-464): 

Equally concerning, however, was the finding that despite many regulatory changes to reduce the discretion of the doctor (26), our study showed that doctors performing the medical assessment for the application continue to have high levels of discretion i.e. doctors decide if a person is sick enough and needy enough to ‘’deserve’’ social assistance, allowing doctors’ opinions and beliefs such as misuse of grants to influence their assessment. Despite perceptions of misuse of grants such as the Disability Grant, Child Support Grant and the Old Age Grant, there is no evidence supporting this. Social grants, however, have been shown to play an important welfare function in poor households as they are typically shared (53).

High levels of discretion in assessments for social grants has previously been shown to be a critical issue in assessing people living with HIV to receive the disability grant (54). The Harmonised Assessment Tool (HAT), developed and piloted jointly by the Department of Social Development and the Department of Health, introduced a more holistic and standardized assessment for the DG, reducing the discretion of the doctor, but has never been implemented. 

3. Could the authors speak to the validity of the TB doctors concern that disability grant funds would be diverted to luxury items rather than nutrition. Did the use of the disability grant come up in their interviews with the patients? Thank you. We have added participants’ views as to what they would use the financial support on to the results section, as well as a few lines in the discussion section on the evidence regarding misuse of grants.

Changes/Additions made to results section (line 352-361):

Participants in the study, however, said they would mostly use the money for food, electricity, rent, and the children’s needs.

“I would be able to give myself energy, buy the foods that are compatible with the treatment. Something that will give me a boost so that I'm alright.” (TB patient, DS-TB, male)

Several participants, mostly women, also mentioned that it would give them back their independence.

“The money helped my sister, because they (household members) received grants but it prioritised their own needs… My mother and my brothers, they received their respective grants. .. but now I can go out and buy what I need as well.” (TB patient, DS-TB, female)

“I'm going to be able to buy it myself and have my own stash (food) at home, where I say okay guys, this cupboard is the sick person’s cupboard.” (TB patient, DS-TB, female)

Addition to discussion (line 452-455): 

Despite perceptions of misuse of grants such as the Disability Grant, Child Support Grant and the Old Age Grant, there is no evidence supporting this. Social grants, however, have been shown to play an important welfare function in poor households as they are typically shared.

REVIEWER 2

1. A little more on cash vs. food support will be useful in putting the things into perspective for future researchers and advocacy. Any studies that have explored that in South Africa will be great value. The common arguments are of misuse of cash (as mentioned by the doctor) or food sharing in case of food support (the recurring theme is food in many quotes). Dear reviewer, thank you for this comment. We have added more information on cash versus food support in South Africa: 1) a paragraph on government provided food parcels (SRD) versus the DG (cash grant) in South Africa to both the results and discussion section, and 2) a few lines on the food versus cash debate in the discussion. 

Changes/Additions made to results section (line 300-305): 

In South Africa, social protection in the form of a Disability Grant (DG) or food parcels, also called Social Relief of Distress (SRD), can support patients in the absence of family but also support families to continue caring for patients. None of the participants in our study, however, received the Social Relief of Distress or food parcel and one of the doctors in the study confirmed that 

“In our experience we find that patients really don’t get that-- Sassa always run out of funds to be able to help patients who require that social relief assistance.” (TB doctor) 

Addition to discussion (line 427-431): 

Effectiveness and efficiency of food versus cash has, to our knowledge, not been investigated in the South African context, yet research conducted in sub-Saharan African, Asian and South-American countries has shown that both cash transfers and food aid are effective but cash transfers and vouchers tend to be more efficient and less costly than food-based interventions. 

2. A point in discussion to emphasize the support to family rather than just the patient. Especially considering the renewed focus on households in National Strategic Plan of South Africa for HIV, TB and STIs 2017-2022. Thank you. We have added a paragraph on the focus on households in the NSP to the discussion. 

Addition to discussion (line 423-427): 

The National Strategic Plan for HIV, TB and STI 2017-2022 recognises the vital role of households in supporting TB patients and the need to empower and strengthen households as key actors. The Department of Social Development is to play the lead role in building strong social support systems and has several tools to its disposal to support TB patients and their households such as the Social Relief of Distress (SRD) or food parcel and the Disability Grant (DG).

3. This was done in 2017-18 and we are now in 2021. It will be a good idea to put this limitation upfront and mention the newer developments or the lack of it, in SDoH/social protection with reference to TB in the country. Thank you. We have added a paragraph to the strengths and limitations section explaining that the data was collected in 2017-2018, nearly 4 years ago. However, we must also make note that this manuscript was submitted to PLOSONE in March 2021. 

Addition to discussion (line 472-478): 

A second limitation is that the interviews were conducted in 2017-2018, nearly 4 years ago. Nevertheless, the challenges presented in this article remain relevant and have to date not been resolved nor addressed. In addition, there has been no change in the social grants policy for TB patients in South Africa.

4. Discuss the findings of this study and lack of evidence for support for patients (nutrition and others) as found by 10.1002/14651858.CD007952.pub3; 10.1002/14651858.CD006086.pub4 Thank you. We included a few lines on the evidence regarding food and/or case support, including the suggested articles (Lutge et al 2015, Grobler et al 2016) as well as more recent evidence (Wingfield et al 2017, Watthananukul et al 2020).

Addition to discussion (line 419-423): 

While there is limited evidence on the effect of food supplementation and financial support on treatment outcomes for TB patients (49, 50), financial support has been shown to positively impact on TB risk factors (16) increase treatment success (17), improve clinic attendance (50) and reduce out-of-pocket payments for TB patients (19).

---

## [Decision Letter · Decision Letter 1]

21 Mar 2022

“I’M SUFFERING FOR FOOD”: FOOD INSECURITY AND ACCESS TO SOCIAL PROTECTION FOR TB PATIENTS AND THEIR HOUSEHOLDS IN CAPE TOWN, SOUTH AFRICA

PONE-D-21-11255R1

Dear Dr. Vanleeuw,

We’re pleased to inform you that your manuscript has been judged scientifically suitable for publication and will be formally accepted for publication once it meets all outstanding technical requirements.

Kind regards,

George Vousden

Deputy Editor-in-Chief

PLOS ONE

Additional Editor Comments (optional):

Your title is currently all in capitals. Please revise this to only use capitals where required. You can find examples of titles here: https://journals.plos.org/plosone/s/submission-guidelines.

Reviewers' comments:

Reviewer's Responses to Questions

**Comments to the Author**

1. If the authors have adequately addressed your comments raised in a previous round of review and you feel that this manuscript is now acceptable for publication, you may indicate that here to bypass the “Comments to the Author” section, enter your conflict of interest statement in the “Confidential to Editor” section, and submit your "Accept" recommendation.

Reviewer #1: All comments have been addressed

Reviewer #2: All comments have been addressed

2. Is the manuscript technically sound, and do the data support the conclusions?

Reviewer #1: Yes

Reviewer #2: Yes

3. Has the statistical analysis been performed appropriately and rigorously? 

Reviewer #1: Yes

Reviewer #2: Yes

4. Have the authors made all data underlying the findings in their manuscript fully available?

Reviewer #1: No

Reviewer #2: Yes

5. Is the manuscript presented in an intelligible fashion and written in standard English?

Reviewer #1: Yes

Reviewer #2: Yes

6. Review Comments to the Author

Reviewer #1: Thank you for addressing my concerns. This paper provides some important insights into entrypoints for interventions aimed at increasing access to TB care and fortifying support for persons with TB.

Reviewer #2: (No Response)

7. PLOS authors have the option to publish the peer review history of their article (what does this mean?). If published, this will include your full peer review and any attached files.

Reviewer #1: No

Reviewer #2: **Yes: **Madhavi Bhargava

---

## [Editor Report · Acceptance letter]

18 Apr 2022

PONE-D-21-11255R1 

“I’m suffering for food”: food insecurity and access to social protection for TB patients and their households in Cape Town, South Africa 

Dear Dr. Vanleeuw:

I'm pleased to inform you that your manuscript has been deemed suitable for publication in PLOS ONE. Congratulations! Your manuscript is now with our production department. 

Kind regards, 

on behalf of

Dr. George Vousden 

Staff Editor

PLOS ONE